# A study of the impact of land transfer decisions on household income in rural China

Peng Wang[1]*, Fanzhi Wang[2]

**1** School of Economics, Lanzhou University, Lanzhou, Gansu, China, **2** School of Finance, Central University of Finance and Economics, Beijing, China

* wangp19@lzu.edu.cn

## Abstract

As an important measure to enhance the allocation of land resources and achieve moderate scale operation in China, land transfer has an irreplaceable role in giving full play to the social and economic value of land. This paper uses data from the China Family Panel Studies in 2018 and applies OLS, PSM and mediator effects to study the impact of land transfer on household income. The results of the study show that: firstly, participation in land transfer can significantly increase the income level of farming households, but the impact of different land transfer acts on household income varies. Secondly, the age and physical condition of the household head, the number of agricultural and non-agricultural members in the household and the poverty status of the household all have a negative effect on household income. On the other hand, the education level of the household head, the household size and the presence of village officials in the household, agricultural subsidies, land titling and land size all have a positive effect on household income. Thirdly, the impact on household income for households that did not participate in land transfer is greater than for those that did. The results of the mediator effect test show that land transfer has a significant effect on household farm income, and that farm income has a full mediator effect, but all have a positive significant effect on household income.

## 1. Introduction

Since the reform and opening up, rural society and systems have been structurally transformed with the development of rural economies, the growth of farmers' incomes, the increase in agricultural yields and the continuous improvement of infrastructure [1–3]. farm land, as the basis for agricultural production and ensuring food security [4, 5], is also an important part of the social-ecological system in many countries. According to FAO's World Agricultural Census in 81 countries (representing 2/3 of the world's population and 38% of the world's land available for cultivation), about 73%, 85% and 95% of farmland is under 1, 2 and 5 hectares respectively [6], which fully indicates the high degree of fragmentation of the world's farmland. In terms of geographical distribution, small-scale farmers are common in Europe, OECD countries and developing countries such as Brazil, India and China. This high degree of farmland fragmentation seriously affects the efficiency of land use and prevents the achievement of large-scale,

**Funding:** This work was supported by the Fundamental Research Funds for the Central

Universities(Grant No.21lzujbkyxs008) and Key R&D Program of Gansu Province, China (Grant No. 20YF3GAO11). The funders had no role in study design, data collection and analysis, decision to publish, or preparation of the manuscript.

**Competing interests:** The authors have declared that no competing interests exist.

mechanized and intensive land management, which in turn reduces the efficiency of land resource use.

At the same time, with the promotion of rural revitalization strategy, land transfer is more and more important to re-allocate land resources. The sustainable use of land resources is the essential and fundamental to improve the economic and social functions of land, which not only changes the rural industrial structure and guarantees the effective supply of agricultural products, but also the rent brought by land transfer reduces the abandonment behavior of farming land of farm households. In addition, the off-farm employment behavior generated by land transfer has an irreplaceable role in increasing household income. Therefore, exploring the impact of arable land on household income is an important topic for the study of rural economic development and for the promotion of agricultural and rural development in countries with small arable land per capita in the world, especially in developing countries [7–9].

## 2. Literature review

With the promotion of urbanization and industrialization in rural areas, the non-farm employment of young and strong laborers has become the main way to improve the household income of farm households, which also causes serious problems of hollowing out and aging in rural areas and the inability of family laborers to support family agricultural production, resulting in serious abandonment of farmland. Land transfer is an important way to improve the efficiency of land resource use and alleviate land abandonment behavior, as well as an important way to the agricultural production difficulties caused by the non-agricultural transfer of labor. Therefore, this paper mainly composes the existing literature from the following aspects.

Firstly, the definition of land titling [10]. The ambiguity of rural land titling reduces the return on farm land transfer while increasing transaction costs, thus lowering the effective supply of land in the transfer market, and the lack of a huge amount of land titling in some developing countries, resulting in the misallocation of land [11]. Besides, the stability and security of land rights help farming households to participate in land transfers [12, 13], while frequent land readjustment policies weaken the stability of land property rights, which in turn hinders the land transfer. In contrast, land titling is the legal empowerment of farming households to clarify their contractual rights to land. Clear land titling helps to enhance the stability and security of land rights, reduce disputes over property rights arising from ambiguity in land titling, as well as information asymmetry between the both parties to the transaction [14], thus improving the efficiency of land use [15] and act on household income.

Secondly, non-farm employment opportunities [15]. With the advancement of both industrialisation and urbanisation, urban non-farm employment opportunities and lucrative labour remuneration have attracted young and strong male labor force in rural areas to migrate to cities [16], which has led to a considerable amount of arable land in the countryside being abandoned and uncultivated while widening the choice of employment for farming households [17]. The majority of farming households are mainly engaged in part-time livelihood options of half-work and half-agriculture [18]. Women and the elderly who cannot support large-scale farming activities due to their health and physical condition have become the labour force of part-time households engaged in farming, while the young and strong male labour force of the household go out to work to earn wage income. This pattern increases the non-farm income [19, 20] of the household and also reduces the dependence of farming households on agricultural activities. As a result, part-time households have a strong willingness to transfer their land, and non-farm employment opportunities are also a major incentive for farming households to move off the land [21].

Thirdly, household characteristics [22], For example, the education level [23], physical condition and age of the household head influence the willingness to transfer farmland [24]. Ali's studies showed that female-headed households are more inclined to rent out land due to the deficiencies of labour force in the household. The younger the household head, the higher the education level, and the greater the non-farm employment capacity, the stronger the willingness to transfer out of the land [25]. Since the existence is usually ensured by activities out of the farm and the maintenance of farming is of secondary importance from economic aspects, so family ideas and farm works factors have a great influence on family behavioral decisions [26]. Factors such as the degree of land fragmentation, mechanised use and economic development of the village all have a significant impact on the decision to transfer land by farming households [27, 28]. In addition, household labour force structure and households with public officials have the advantage of capturing information about transactions, thus increasing the opportunities of land transaction [29].

Finally, the environment for economic development [30]. The experience of many transition countries proves that circular economy can raise the cost-effectiveness in all economies and get larger economic benefits [31]. With the implementation of the strategies of poverty alleviation and rural revitalisation, the demand for rural development has given rise to new agents for agricultural management [32], which requires a mass of land for large-scale and specialised production [33], and can even directly drive scattered farmers towards new agents for agricultural management. Hence, this kind of moderate scale operation has a demonstration and driving effect on surrounding farmers, which accelerates the land transfer and increases the economies of scale. In contrast, new agricultural operators have strong capital to purchase large agricultural machinery, and provide mechanized services [34, 35], guidance on agricultural skills and products acquisition to other farmers. In this sense, not only are the natural and market risks of farming reduced through capital and technology spillovers, but the income level, production and management capacity, and marginal agricultural output rate of farming households are also increased, which effectively drives the demand for land transfer [36].

Compared to existing studies, the marginal contributions of this paper are as follows. (i) In terms of research perspective, this paper pays more attention to the income effects of land transfer. Considering that official data is difficult to quantify the behavioural choices made by farmers on land transfer, micro data from the China Family Panel Studies in 2018 are used to examine the impact of land transfer on farmers' income, which largely improves the accuracy and feasibility of the research results. (ii) With respect to the research content, this paper explores the heterogeneous impact of land transfers in groups, while taking into account the differentiation between groups of farmers, thus providing a micro-foundation for rural land reform. (iii) As for research methodology, to address the endogeneity problem caused by missing variables and sample selection, regression analysis is conducted using OLS, PSM and mediator effects, which further enhances the reliability of the findings.

The rest of this paper is organized as follows. Section 3 introduces data sources, model construction. and variable selection. Section 4 contains the empirical results and robustness tests. In Section 5, the mechanisms of land transfer and farm income on household income are discussed. Finally, Section 6 draws the conclusions of the study.

## 3. Data, variable selection and descriptive statistics

### 3.1. Data

The data used in this study are from the China Family Panel Studies (CFPS) which is a nationally representative, annual longitudinal survey sponsored by the Institute of Social Science Survey, Peking University. CFPS collected data of 25 provinces, multiple levels of representation,

**Table 1. Number of matched samples in each group.**

|  | Land transfer | Land inflow | Land outflow |
|---|---|---|---|
| Untreated | 5981 | 5966 | 5914 |
| Treated | 1928 | 1437 | 493 |
| Total | 7909 | 7403 | 6407 |

on a rural, urban, provincial, and national scale. The survey data is comprehensive and informative, covering various aspects such as the basic structure of rural households, income and expenditure, agricultural production and operation, and land use and transfer, etc. In order to explore the heterogeneity of land transfer on farm households, this paper, based on data cleaning, retained key variables such as land transfer, land certificate, household income, individual characteristics of household head and household characteristics, and finally obtained the tracking data for a sample of 7,913 rural households in 2018, including 6475 households are in the land flow-out group (Including 5982 non-flow households), 7420 households are in the land flow-in group (Including 5982 non-flow households). The number of matched samples in each group is shown in Table 1. Compared with similar studies in the past, the data in this paper are more widely distributed and the sample size is larger, which makes it more representative.

## 3.2. Research methods

The most intuitive approach to study the impact of land transfer on farm household income is to compare the income of households involved in land transfer (treatment group) with that of households not involved in (control group). But several issues need to be considered in the use of this approach. Firstly, farming households have to face a 'self-selecting' behaviour, where the household decision-maker may be influenced by life circumstances, household endowments, and many uncontrollable factors, thus making different decisions. Secondly, when looking at the income of a household that has participated in land transfer, it is not possible to keep abreast of the household income of a household that has not participated in land transfer; to the contrary, it is not possible to look at that of a household that hasn't done the same when looking at the income of a household that has not participated in land transfer. Thus, they face a "counterfactual problem" caused by "missing data".

In view of this, the propensity score matching (PSM) model chosen for this paper reduces the bias in the empirical results caused by the data to the greatest extent possible. The following model was therefore constructed to estimate the impact of land transfer on household income.

$$InY_{ij} = \beta_0 + \sum \beta_m hc_{ij} + \sum \beta_k fc_{ij} + \sum \beta_n control_{ij} + \varepsilon \tag{1}$$

In Eq (1), $InY_{ij}$ denotes the logarithm of the farm household's income (where, the subscript $ij$ indicates the household's land transfer status; 'j = 1' for 'have land transferred', 'j = 0' for 'have no land transferred'), $hc$ denotes the control variable for the characteristics of the household head; $fc$ denotes control variables for household characteristics; $control$ denotes other control variables; $\beta_0$, $\beta_m$, $\beta_k$ and $\beta_n$ are the estimated parameters for each variable, and $\varepsilon$ is a random error term.

$$ATT = E(InY_{i1} - InY_{i0}|j = 1, X = x \tag{2}$$

$$ATU = E(InY_{i1} - InY_{i0}|j = 0, X = x) \tag{3}$$

In Eqs (2) and (3), $InY_{ij}$ denotes the logarithm of the income of farm households $i$

participation in land transfer, $InY_{i0}$ denotes the logarithm of the income of farm household $i$ not participation in land transfer. $ATT$ denotes the average treatment effect of farm households that have land transferred, $ATU$ denotes the average treatment effect for farm households that have no land transferred.

## 3.3. Variable selection

Reasonable variable selection plays an important role in improving the robustness and universal applicability of research results. This paper focuses on the impact of agricultural land transfer on household income, taking into full consideration the actual situation of the sample households, thus making the choice of variables relevant.

**3.3.1. Dependent variables.** In this paper, the logarithm of the total income of the sample households is used as the dependent variable. As income and land are important to guarantee the sustainable development of farming households, how to broaden the income channels of rural households and raise the income level of farmers has become one of the most crucial concerns for many scholars and policy makers. This paper examined the impact of land transfer on household income from the perspective of land transfer, which promotes the optimization of farmers' family resources, improves household income levels, and even lays a micro-foundation for rural land system reforms.

**3.3.2. Independent variables.** The three variables of whether a farm household is involved in land transfer, whether the farm has transferred land into the household and whether the farm has transferred land out of the household are used as the independent variables in this study. In the questionnaire, the item that "At present, the right to operate your household's farmland has been transferred to agency " was selected as the land outflow group, and the item that " Whether your family is involved in land transfer at present " was selected as the land inflow group. The assignment 1 indicates whether the farming household is involved in land transfer is bringing land inflows and land outflows. The assignment 0 indicates farming households not involved in land transfer. For farming households, land has both economic and social value, with the economic value being mainly reflected in the business income that farming provides to farming households, and the social value being mainly as an important guarantee for the continuation of household development. When there is a large-scale transfer of land, their economic conditions, lifestyles and feelings change significantly. Therefore, the participation of farming households in land transfer is an optimal household decision made after careful consideration [37].

**3.3.3. Control variables.** There is a cohort effect on whether a rational farmer makes a decision to transfer land. To make the model estimation more accurate, household head characteristics, household demographic characteristics and other control variables were introduced. As the main decision maker of a household [38, 39], the household head has an important role in the household development and an influence on the household members. As a consequence, age, education level and physical condition were included in the characteristics of household head. In terms of demographic characteristics of households, this paper focuses on variables such as the size of farming household [40], the number of people engaged in agricultural production activities, the number of people engaged in non-agricultural production activities and the presence or absence of village officials in the household. Among the other control variables in terms of household, affluence, has obtained a land management right certificate [41], and receives agricultural subsidies, the size of household land was used to represent the other control variables of a household, as shown in Table 2.

**Table 2. Variable definitions and descriptive statistics.**

| Variables | Definitions | Land transfer | | Land inflow | | Land outflow | |
|---|---|---|---|---|---|---|---|
| | | *Mean* | *Std* | *Mean* | *Std* | *Mean* | *Std* |
| Income | Total household income(in the form of natural logarithm) | 9.173 | 2.620 | 9.140 | 2.662 | 9.146 | 2.614 |
| Land transfer | 1 will be assigned if households participate in land transfer, otherwise it is 0 | 0.244 | 0.430 | 0.194 | 0.395 | 0.076 | 0.265 |
| Age | Age of the household head (years) | 54.663 | 11.058 | 54.415 | 11.020 | 55.007 | 11.172 |
| Education | Education of the household head: 1 = Illiteracy; 2 = Primary School; 3 = Junior School; 4 = High School; 5 = Vocational High School; 6 = Higher Education; 7 = Undergraduate | 2.508 | 0.929 | 2.509 | 0.930 | 2.508 | 0.939 |
| Health | Physical condition of the household head:1 = Fine; 2 = Well; 3 = General; 4 = Bad; 5 = Poor | 2.812 | 0.984 | 2.805 | 0.992 | 2.817 | 0.983 |
| Household size | Total number of family members | 4.260 | 1.884 | 4.281 | 1.884 | 4.245 | 1.897 |
| Number of farmers | Number of agricultural labour forces in a household Characteristics of household | 1.956 | 0.900 | 1.958 | 0.897 | 1.961 | 0.915 |
| Number of workers | Number of workers in a household Characteristics of household | 1.224 | 0.925 | 1.243 | 0.919 | 1.198 | 0.933 |
| Village officials | 1 will be assigned if a family member has serves on the village council, otherwise it is 0 | 0.062 | 0.241 | 0.062 | 0.241 | 0.061 | 0.239 |
| Poor households | 1 will be assigned if a household is poor, otherwise it is 0 | 0.155 | 0.362 | 0.156 | 0.363 | 0.158 | 0.365 |
| Land certificates | 1 will be assigned if a family has land operation certificate, otherwise it is 0 | 0.416 | 0.493 | 0.409 | 0.492 | 0.410 | 0.491 |
| Subsidies | 1 will be assigned if a family has agricultural subsidies, otherwise it is 0 | 0.707 | 0.455 | 0.705 | 0.456 | 0.697 | 0.459 |
| Land size | Household's cultivated land area (mu[a])(in the form of natural logarithm) | 1.813 | 1.118 | 1.805 | 1.124 | 1.772 | 1.086 |
| Observations | | 7913 | | 7420 | | 6475 | |

[a] 1 mu = 1/15 hectare

# 4. Empirical results and discussion

## 4.1. Results of the OLS model estimation

Table 3 shows that participation in land transfer, land inflow and land outflow play a positive and significant role in household income compared to farming households that do not do the same. Jin shows that both land outflow and land inflow to farming households can increase household income [21], while some scholars argued that only land transfer can increase the marginal output rate of land and achieve large-scale operation, thus promoting agricultural technology and increasing farming household income [42].

Householder characteristics. As the main decision maker of households, the household head has a subtle influence on the behavioural choices of the family members. As the age of household head increases, he/she has become more experienced in agricultural production, which is conducive to increasing the household income. But when the age reaches a certain value, the physical condition of a person also decreases, and the non-agricultural employment opportunities decrease; coupled with the heavy but less profitable agricultural production and high natural and market risks, unfavourable increase in the household income will be seen. With the widespread application of science and technology, agricultural machinery and scientific farming have made agricultural production activities relatively complex [43], which requires a higher level of education for the workers. In this sense, the higher the level of education of a household head, the more receptive he is to new things, which enables him to grasp development opportunities in time and helps to increase the household income. Table 2 shows that the higher the education level of a household head, the more land transfer has a positive and significant effect on household income.

Family demographics. Based on the theory of internal division of labour within a household, if the size of the household is larger, it will be more likely to help to refine the division of household members. Engaging in diversified production not only improves the allocation

**Table 3. Estimation results of the OLS regression.**

| Variables | Land transfer | Land inflow | Land outflow |
|---|---|---|---|
| Land transfer | 0.260*** | | |
| | (0.063) | | |
| Land inflow | | 0.135* | |
| | | (0.075) | |
| Land outflow | | | 0.621*** |
| | | | (0.083) |
| Age | -0.006* | -0.007* | -0.008** |
| | (0.004) | (0.004) | (0.004) |
| Education | 0.324*** | 0.321*** | 0.318*** |
| | (0.031) | (0.032) | (0.033) |
| Health | -0.110*** | -0.114*** | -0.104*** |
| | (0.029) | (0.031) | (0.032) |
| Household size | 0.275*** | 0.279*** | 0.277*** |
| | (0.015) | (0.016) | (0.016) |
| Number of farmers | -0.014 | -0.020 | -0.009 |
| | (0.030) | (0.032) | (0.031) |
| Number of workers | -0.113*** | -0.124*** | -0.134*** |
| | (0.041) | (0.043) | (0.045) |
| Village officials | 0.451*** | 0.446*** | 0.411*** |
| | (0.097) | (0.102) | (0.109) |
| Poor Households | -0.274*** | -0.283*** | -0.282*** |
| | (0.074) | (0.077) | (0.081) |
| Subsidies | 0.343*** | 0.359*** | 0.299*** |
| | (0.069) | (0.072) | (0.075) |
| Land titling | 0.165*** | 0.143** | 0.168*** |
| | (0.056) | (0.059) | (0.062) |
| Land size | 0.040 | 0.046 | 0.064* |
| | (0.031) | (0.032) | (0.035) |
| Constant | 7.584*** | 7.624*** | 7.688*** |
| | (0.284) | (0.297) | (0.311) |
| Observations | 7913 | 7420 | 6475 |
| R-squared | 0.073 | 0.071 | 0.074 |
| Root MSE | 2.525 | 2.568 | 2.517 |

Note: Those in parentheses are robust standard errors, where *** $p<0.01$, ** $p<0.05$, and * $p<0.1$, respectively

efficiency of the household's human resources, but also increases the household's ability to withstand risks while widening the channels of household income, which in turn acts on household income. Hence, the larger the household size, the more it plays a positive and significant role in household income [44]. Both agricultural and non-agricultural labour forces in the household have a negative effect on household income. Whether it is the land inflow group or the land outflow group, most rural households mainly take part-time household employment as the best livelihood choice, when the number of non-agricultural labour force in the farming household is more than that of agricultural labour force. In this case, the non-farm income can compensate for the income from agricultural production, thus creating a substitution effect. Conversely, when the number of agricultural labour force in the farming household is greater than that of non-agricultural labour force, the farm income will also compensate for

**Table 4. Balance test of explanatory variables before and after matching.**

| Variables | Matching methods | Ps R2 | LR chi2 | Mean Bias | Med Bias | B | R |
|---|---|---|---|---|---|---|---|
| Land transfer | Before matching | 0.011 | 96.37 | 5.9 | 2.8 | 25.9 | 0.98 |
| | Nearest neighbour matching | 0.001 | 3.46 | 1.5 | 1.3 | 6.0 | 1.02 |
| | Radius matching | 0.002 | 6.47 | 1.5 | 0.7 | 8.2 | 1.13 |
| | Spline matching | 0.001 | 7.99 | 2.3 | 2.2 | 9.1 | 0.98 |
| Land inflow | Before matching | 0.014 | 101.76 | 8.1 | 5.6 | 29.7 | 1.02 |
| | Nearest neighbour matching | 0.001 | 2.25 | 1.1 | 0.7 | 5.6 | 0.86 |
| | Radius matching | 0.001 | 5.59 | 1.7 | 1.0 | 8.8 | 1.21 |
| | Spline matching | 0.000 | 1.34 | 0.9 | 0.8 | 4.3 | 1.03 |
| Land outflow | Before matching | 0.029 | 102.21 | 13.8 | 10.4 | 48.0 | 0.95 |
| | Nearest neighbour matching | 0.001 | 1.77 | 2.4 | 2.0 | 8.5 | 0.73 |
| | Radius matching | 0.007 | 9.25 | 5.3 | 3.2 | 19.4 | 1.07 |
| | Spline matching | 0.005 | 6.90 | 3.7 | 3.7 | 16.7 | 0.96 |

the non-farm income. Thus, there will be a substitution effect between farm income and non-farm income which substitution effect does not increase the income of the farming household. Farming households with village officials tend to be responsive to development opportunities and information, and have a degree of discretion to intervene in land transfers in accordance with management needs and interests. The empirical results showed that farming households with land outflow are able to take advantage of more development opportunities to raise household income.

Affluence, receipt of agricultural subsidies, land titling, and the size of the household arable land, were taken as other control variables. Poverty attribute plays a negative and significant role in household income for both the full sample and the land inflow and land outflow groups. This is because poor households suffer a serious shortage of development resources and, when allocating land for transfer, they may face adjustments in their development strategies, breaking the previous development rhythm and leading to a decline in household income. Agricultural subsidy is an incentive guarantee for households to engage in agricultural production activities [45]. Table 4 demonstrates that land inflow contributes 6% or more to household income than land outflow, as household inflow of land expands the scale of household cultivation [28]. Households then increase production efficiency through continuous mechanized production, which in turn reduces the cost of agricultural production. Coupled with the vigorous support of China for agricultural development and a considerable amount of inclusive transfers out in favour of new agricultural operators operating on a moderate scale, the cost of agricultural production is reduced while increasing the income level of farm households. Land titling has a positive and significant effect on the income of land-transferring households, with land titling increasing non-agricultural employment, and the farm households' willingness to transfer land [14, 30]. Thus, the household income level can be raised. But some studies revealed that land titling has a dampening effect on increasing household income [46]. Household land size has a positive but insignificant effect on household income in the land transfer and land inflow groups, while households in the land outflow group have a positive and significant effect on household income at the 10% statistical level.

## 4.2. Balanced hypothesis testing

To better verify the robustness of the regression results, this paper drew on studies of Rubin to re-estimate the impact of land transfer on farm household income using the PSM method [47]. Balance tests were conducted in terms of Ps R2, LR chi2, Mean Bias, Med Bias B and R.

Firstly, the standard deviations of the matched variables between the treatment and control groups after matching were examined; a decrease in the standard deviations indicated a decrease in the difference between the two groups. A standard deviation of less than 20% between the matched samples of the treatment and control groups [48] means that the matching is more successful. Secondly, this study further examined whether there is a difference between the means of the matching variables in the treatment and control groups, and determined the significance of the difference using a robustness test. Thirdly, the Pseudo-R2, $\chi^2$, Mean bias, B values and R values were examined to test whether the matching met the equilibrium assumption overall. Among other things, B is Rubin's B, the standard deviation in PS means between the treatment and control groups, and R is Rubin's R, the ratio of PS variances between the treatment and control groups. If B<25% and R falls within the scope of [0.5, 2] [44], the matching balance assumption is considered to be fully satisfied.

In addition, three matching methods such as Nearest Neighbour Matching (1:4), Radius Matching, and Kernel Matching, were used in this paper to test the matching balance assumptions for land transfer, land inflow, and land outflow, respectively, which results for each group are shown in Table 4. Compared to the pre-matched results, most of the standard biases were reduced after matching and all of the standard biases were 10%. Further, Pseudo-R2, $\chi^2$, mean bias, B-values and R-values were reduced after matching compared to the pre-matched results, with all B-values less by 25% and R-values ranging from 0.95 to 1.21. The results illustrate that the samples were matched more perfectly.

## 4.3. Conditions of common support regions

The conditions of common support regions reject the tails of the PSM distribution to improve the matching quality, and the non-parametric approach is only meaningful if built on a common support domain [48]. The credibility of the matching results is mainly reflected in whether the common support regions between the treatment and control groups satisfies the size of the overlap region between the two decision behaviours under the influence of the same control variables. When the overlap of the common support regions between the treatment and control groups is great, it indicates that the credibility of the study results is effectively improved after the samples are matched, thus lowering the matching precision due to sample loss. In order to observe more visually the differences in the propensity score values of the treatment and control groups before and after matching, the corresponding kernel density function plots were plotted separately for each group, each of which shows that the conditions of common support regions were satisfied after matching (See Figs 1–3).

As Table 5 shows, in the land transfer group, specifically, participation in land transfer has a positive and significant effect on household income at the 1% statistical level, with ATT and ATU values of 0.286 and 0.299, respectively. In addition, the contribution of participation in land transfer to raising the household income of farmers is 28.6%, while the household income of non-participating households would have been 29.9% higher if they had participated in land transfer. Possible reasons for this are as follows. First, when farm households participate in land outflows, they achieve both the optimal allocations of household land resources and break the seasonal constraints of agricultural production. These further enable them to search for development information and build social capital and networks through various channels to enhance employment opportunities. Secondly, when farm households participate in land inflow, it will encourage farm households to increase the degree of mechanization [49] to reduce production costs and increase the economic benefits of large-scale land management if the land inflow is concentrated or the area of land is large, which in turn will contribute to the increasing of household income.

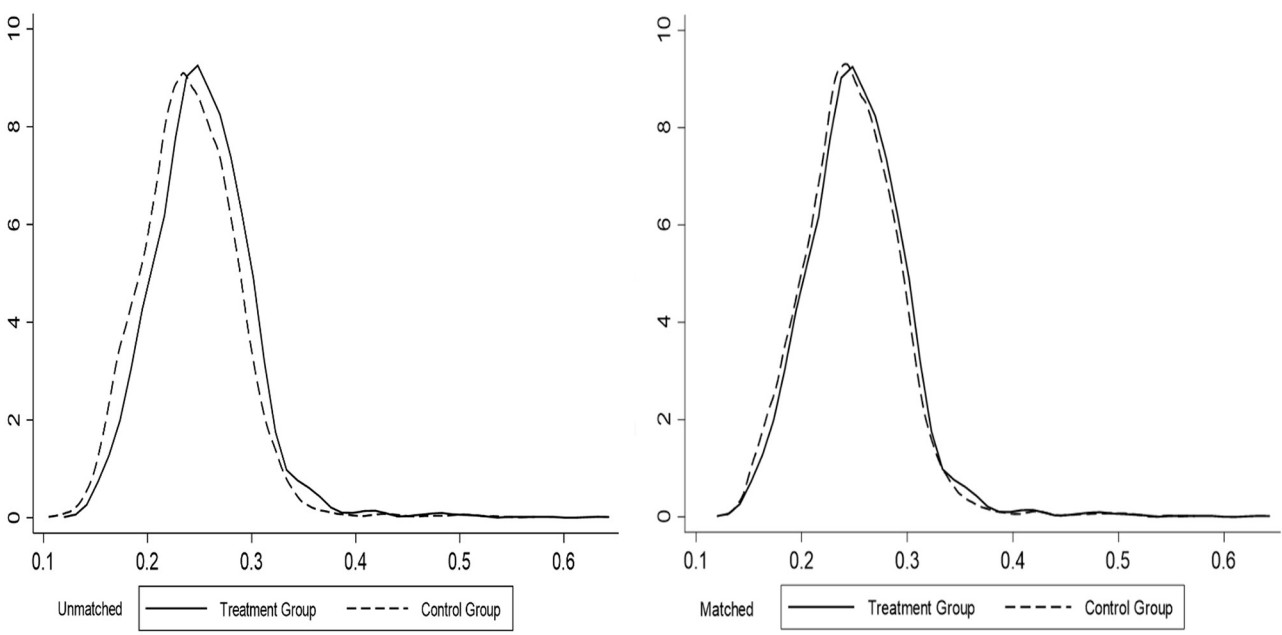

**Fig 1. Common support regions between land transfer group and land non-transfer group.**

In the land transfer group, participation in land inflow has a positive and significant effect on household income at the 5% statistical level, where the values of ATT and ATU are 0.163 and 0.178, respectively. What's more, the contribution of participation in land inflow to raising the household income of farmers is 16.3%, while the household income of non-participating

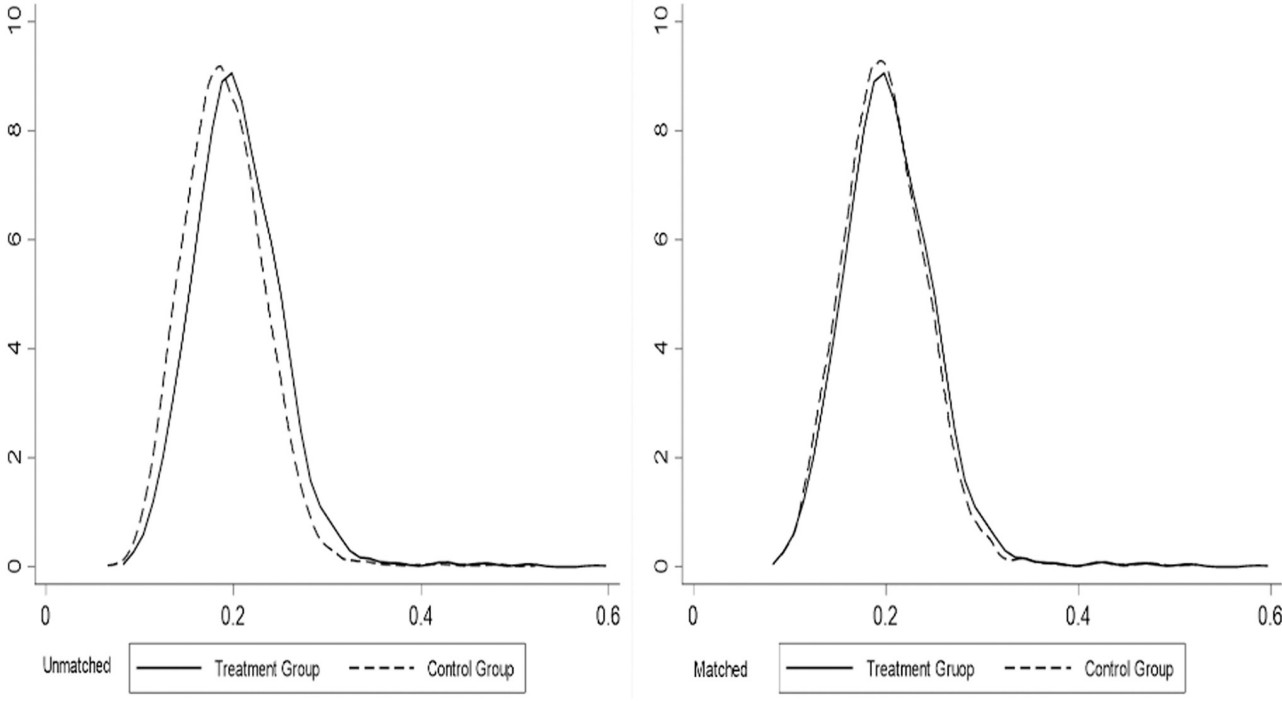

**Fig 2. Common support regions between flow-in group and land non-transfer group.**

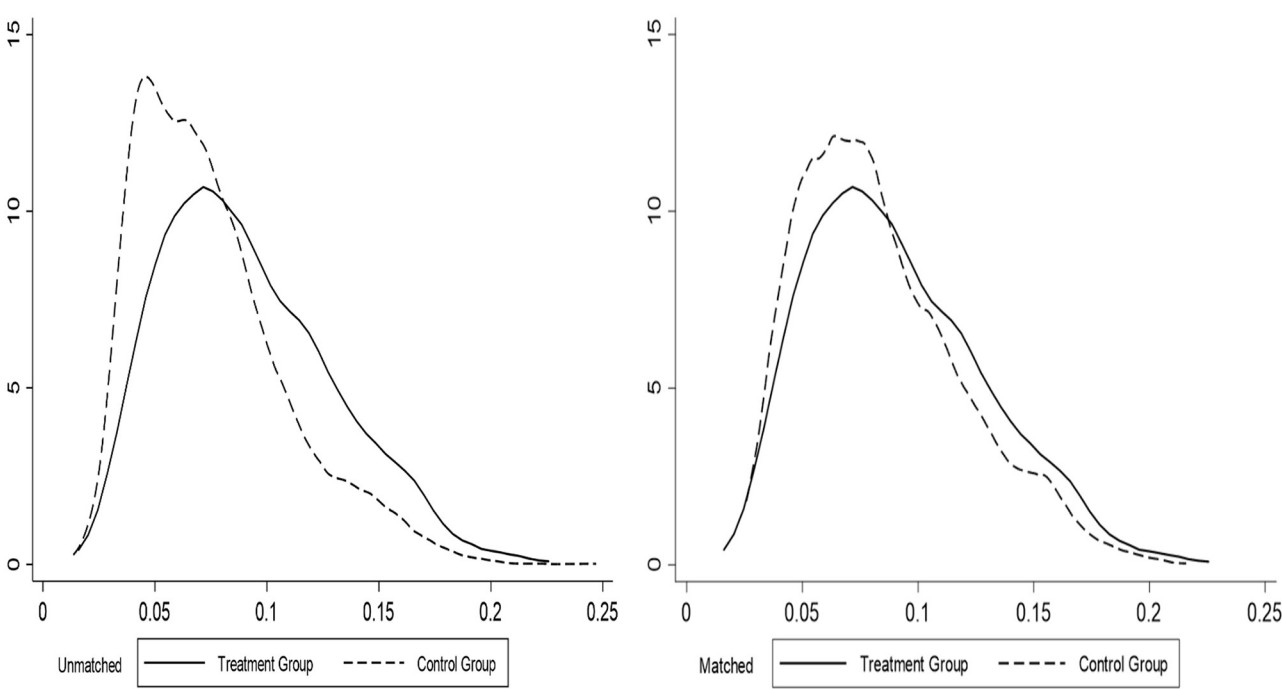

**Fig 3. Common support regions between flow-out group and land non-transfer group.**

households would have been 17.8% higher if they had participated in land inflow. Possible reasons for this are as below. Firstly, land inflow will expand the size of farming land for households and promote moderate scale production for farming households while increasing crop yields through mechanized production [50], and scientific cultivation, which will in turn act on household income. Secondly, farmers will ensure the stability of farming income by alleviating the farming risks brought about by nature and the market through diversification of cultivation. In addition, land inflow households are subject to high input costs in hiring and

**Table 5. Results of the PSM model estimation.**

| Matching methods | | | Land transfer | Land inflow | Land outflow |
|---|---|---|---|---|---|
| Nearest neighbour matching | ATT | | 0.286*** | 0.122 | 0.646*** |
| | | | (0.089) | (0.103) | (0.150) |
| | ATU | | 0.299*** | 0.142 | 0.748*** |
| | | | (0.079) | (0.091) | (0.104) |
| Radius matching | ATT | | 0.267*** | 0.163** | 0.572*** |
| | | | (0.065) | (0.076) | (0.084) |
| | ATU | | 0.288*** | 0.178** | 0.621*** |
| | | | (0.065) | (0.078) | (0.085) |
| Spline matching | ATT | | 0.248*** | 0.129* | 0.630*** |
| | | | (0.065) | (0.077) | (0.086) |
| | ATU | | 0.258*** | 0.125 | 0.684*** |
| | | | (0.068) | (0.077) | (0.091) |

Note: Due to space constraints and positive effect of the regression results, only Radius matching was reported in this paper. Those in parentheses are robust standard errors, where *** $p<0.01$, ** $p<0.05$, and * $p<0.1$, respectively.

production factor while expanding their farming operations. For this reason, they are unable to earn wage income, making the income effect of land inflow households lower than that of land outflow households.

In the land outflow group: participation in land outflow has a positive and significant effect on household income at the 1% statistical level, where the values of ATT and ATU are 0.572 and 0.621, respectively. Besides, the contribution of participation in land outflow to raise the household income of farmers is 57.2%, while the household income of non-participating households would have been 62.1% higher if they had participated in land outflow. Possible reasons for this are as follows. First above all, when a farm household loses its land, it shifts some of its young and middle-aged labour force to the non-agricultural sector to earn wage income [51], but it also loses farm income, and there is a substitution effect between wage income and farm income. But the impact of wage income on total household income is much greater than farm income. Secondly, women and the elderly are the only ones who remain in the households that have lost their land. This group of people are once again involved in agricultural production in the form of hired labour, earning a wage income without bearing the risks of farming, which also alleviates the shortage of land for the realization of large-scale production in rural areas.

In summary, the income of farming households involved in land transfer is higher than that of households not involved in. If farming households not involved in land transfer are involved in, the change in household income is greater than that of farming households involved in land transfer, with the effect of income gained by farming households out of land being the greatest. It is clear that participation in land transfer has a significant impact on increasing household income, regardless of whether the household is moving into or out of the land.

## 5. Impact of land transfer and farm income on household income

This paper draws on the mediator effects test proposed by Baron [52] to test the mechanism of action of how land transfer affects household income through farm income. The mediator effects model is constructed as follows.

$$Income = \alpha_0 + \alpha_1 Transfer + \alpha_i X_i + \varepsilon_1 \tag{4}$$

$$Aincome = \beta_0 + \beta_1 Transfer + \beta_i X_i + \varepsilon_2 \tag{5}$$

$$Income = \gamma_0 + \alpha_1' Transfer + \gamma_1 Aincome + \gamma_i X_i + \varepsilon_3 \tag{6}$$

In Eqs (4), (5) and (6), *Income* represents total household income, *Aincome* represents the household farm income, a mediating variable, $X_i$ represents a control variable that affects both *Income* and *Aincome*. Moreover, $\varepsilon_1, \varepsilon_2, \varepsilon_3$ are random error terms. $\alpha_0, \beta_0, \gamma_0$ denote a constant term; $\alpha_1, \beta_1, \gamma_1$ are the coefficients of the independent variables. $\alpha_1$ in Eq (4) is the effect of *Transfer* on *Income*, $\beta_1$ in Eq (5) is the effect of *Transfer* on *Aincome*, $\alpha_1'$ and $\gamma_1$ in Eq (6) are the effects of *Transfer* on *Aincome* on *Income*, respectively. Only when $\alpha_1$ in Eq (4) is significant can we proceed to test the significance of $\beta_1, \alpha_1'$ and $\gamma_1$. In the case where $\alpha_1$ is significant, and both $\beta_1$'s in Eq (5) and $\alpha_1$'s in Eq (6) are significant, it accounts for that at least part of the effect of *Transfer* on *Income* is achieved through *Aincome*. When $\alpha_1'$ in Eq (6) is significant, *Aincome* plays a partially mediator effect, otherwise it has a fully mediator effect.

Table 6 reports the test results for the mediator effect of land transfer on household income. Columns 1, 4 and 7 show the effect of land transfer, land inflow, and land outflow on household income, respectively, corresponding to $\alpha_1$ in Eq (4). Columns 2, 5 and 8 exhibit the effect

**Table 6. Regression results of intermediary effect.**

| Variables | Land transfer | | | Land inflow | | | Land outflow | | |
|---|---|---|---|---|---|---|---|---|---|
| | Income | Farm income | Income | Income | Farm income | Income | Income | Farm income | Income |
| Land transfer | 0.260*** | 0.363*** | 0.185*** | | | | | | |
| | (0.063) | (0.121) | (0.058) | | | | | | |
| Land inflow | | | | 0.135* | 0.735*** | -0.024 | | | |
| | | | | (0.075) | (0.137) | (0.068) | | | |
| Land outflow | | | | | | | 0.621*** | -0.728*** | 0.766*** |
| | | | | | | | (0.083) | (0.205) | (0.083) |
| Farm income | | | 0.208*** | | | 0.217*** | | | 0.200*** |
| | | | (0.006) | | | (0.007) | | | (0.007) |
| Control variable | Yes | Yes | Yes | Yes | Yes | Yes | Yes | Yes | Yes |
| Observations | 7913 | 7913 | 7913 | 7420 | 7420 | 7420 | 6475 | 6475 | 6475 |
| Pseudo R2 | 0.073 | 0.050 | 0.198 | 0.070 | 0.052 | 0.203 | 0.074 | 0.046 | 0.188 |

Note: Those in parentheses are robust standard errors, where ***$p<0.01$, **$p<0.05$, and *$p<0.1$, respectively

of land transfer, land inflow and land outflow on household farm income, respectively, corresponding to $\beta_1$ in Eq (5). Columns 3, 6 and 9 demonstrate the effect of the farm income, a mediator variable, on household income, respectively, corresponding to $\gamma_1$ in Eq (6). However, the regression results for land transfer in column 6 are insignificant and the corresponding $\alpha_1'$ in Eq (6) is insignificant, meaning that *Aincome* has a full mediating effect on *Income*. Judging from the regression results, the coefficient estimates of all other variables are significant, stating that the mediator effect of farm income exists and is a partial mediator effect to some extent.

Firstly, participation in land transfer has a positive and significant effect on both household income and farm income at the 1% statistical level, increasing by 26% and 36.3% respectively. When both mediator variables, farm income and land transfer, were included in the regression model, the effect of land transfer on household income was reduced by 7.5%. The possible reason for this is that if a household participates in land transfer, it will face a transformation of the household development structure, breaking the original development model and leading to a period of transition pain for the household; in the end, this makes land transfer less of an impact on household income.

Secondly, land inflow has a positive and significant effect on household income and farm income at the 10% and 1% statistical levels, respectively, especially the contribution of land inflow to raising household farm income is 73.3%. But when both mediator variables, farm income and land inflow, were included in the regression model, the significance of land inflow on household income disappeared, which is probably because when households transfer to land, although they can expand the scale of household business. But it is possible that when households transfer land, they are able to expand their farming operations but subject to the impact of both natural and market risks, and the high cost of production and management makes the net farm income lower.

Finally, land outflow has a negative and significant effect on household farm income, reducing it by 72.8%. But when both mediating variables, farm income and land outflow, were included in the regression model, the contribution of land outflow to household income is 76.6%. On the one hand, when households transfer out of land, the reduction in household acreage can have a significant impact on household farm income afterwards. On the other hand, some households will change their traditional farming practices and adopt a scientific

approach to land farming in order to increase marginal output rates, while young and middle-aged household labourers will shift to seek for non-agricultural employment opportunities and earn wage income, thus increasing household income levels.

## 6. Conclusions and policy implications

### 6.1. Conclusions

Firstly, the act of land transfer has a significant impact on the household income of farmers, but the impact of different acts of land transfer on household income varies, suggesting that participation in land transfer and land inflow have a positive and significant impact on household income, while land outflow has the greatest impact on household income. On the one hand, land transfer breaks the traditional development model of households, furthers the transformation of household development structure and achieves the optimal allocation of household resources. On the other hand, the act of land transfer widens the income channels of households, thus affecting household income.

Secondly, household head and household characteristics affect both land transfer behaviour and income structure of farming households. The study shows that the age and physical condition of the household head plays a negative and significant role in household income, while the education level of the household head plays a positive and significant role in household income. The total number of household members and the presence of village officials in the household play a positive and significant role in household income, but the number of farm and non-agricultural members in the household play a negative role in household income. Among the other control variables, agricultural subsidies, land titling and land size all play a positive role in household income, except for affluence.

Thirdly, the PSM analysis verifies that the impact on household income is greater for households that did not participate in land transfer than for those that have done the same. The results of the mediator effect test elaborated that land transfer has a significant effect on household agricultural income, among which that of land outflow is negative. When both agricultural income and land inflow are included in the regression model, the significance of land inflow on household income disappears, indicating that agricultural income has a complete mediator effect, but agricultural income has a positive significant effect on household income.

### 6.2. Policy implications

Firstly, the willingness of farming households to transfer their land and regional differences should be fully respected, and different types of farming households should be guided by categories, so as to avoid the negative impact of blind transfers on farming households. Secondly, relevant authorities should continue to intensify land transfer policies, strengthen the dynamic management of land transfer and carry out innovations in regional land transfer models, so as to balance the interests between transfer agents and farmers on the basis of synergistic policy effects. Thirdly, measures should be taken to consolidate agricultural skills training for farmers and raise agricultural subsidies, so as to avoid land abandonment and scarcity while achieving moderate scale farming by farmers, thereby increasing family income and the efficiency of land use. Last but not the least, it is necessary to actively cultivate the rural land transfer market, strengthen the registration of rural land rights and titles, enhance the management transparency of land transfer and the awareness of transfer contracts among farming households, and lower the transaction costs in the process of land transfer.

There are many studies on the distribution of land transfer on household income, and due to the limitation of data availability, this paper only uses the data of CFPS 2018. Although this data has wide coverage and strong representation, this paper does not use panel data to study

the impact of land transfer on household income, which cannot reflect the long-term variability of farm household income. This is a direction for future research to track changes in farm household income over time.

## Supporting information

**S1 File.**
(DTA)

**S2 File.**
(DTA)

**S3 File.**
(DTA)

## Author Contributions

**Data curation:** Peng Wang.

**Formal analysis:** Peng Wang.

**Methodology:** Peng Wang.

**Writing – original draft:** Peng Wang, Fanzhi Wang.

**Writing – review & editing:** Peng Wang, Fanzhi Wang.

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
