## [Decision Letter · Decision Letter 0]

22 Aug 2022

PONE-D-22-16858A study of the impact of land transfer decisions on household income in rural ChinaPLOS ONE

Dear Dr. Wang,

Thank you for submitting your manuscript to PLOS ONE. After careful consideration, we feel that it has merit but does not fully meet PLOS ONE’s publication criteria as it currently stands. Therefore, we invite you to submit a revised version of the manuscript that addresses the points raised during the review process.

We look forward to receiving your revised manuscript.

Kind regards,

László Vasa, PhD

Academic Editor

PLOS ONE

Journal Requirements:

2. Thank you for including your ethics statement:  "N/A". For studies reporting research involving human participants, PLOS ONE requires authors to confirm that this specific study was reviewed and approved by an institutional review board (ethics committee) before the study began. Please provide the specific name of the ethics committee/IRB that approved your study, or explain why you did not seek approval in this case.

4. Please ensure that you include a title page within your main document. We do appreciate that you have a title page document uploaded as a separate file, however, as per our author guidelines (http://journals.plos.org/plosone/s/submission-guidelines#loc-title-page) we do require this to be part of the manuscript file itself and not uploaded separately.

6. We note you have included a table to which you do not refer in the text of your manuscript. Please ensure that you refer to Tables 4& 5 in your text; if accepted, production will need this reference to link the reader to the Table.

7. Please ensure that you refer to Figure xxxxx in your text as, if accepted, production will need this reference to link the reader to the figure

8. Please include a separate caption for each figure in your manuscript.

Reviewers' comments:

Reviewer's Responses to Questions

**Comments to the Author**

1. Is the manuscript technically sound, and do the data support the conclusions?

Reviewer #1: Partly

Reviewer #2: Yes

2. Has the statistical analysis been performed appropriately and rigorously? 

Reviewer #1: Yes

Reviewer #2: Yes

3. Have the authors made all data underlying the findings in their manuscript fully available?

Reviewer #1: Yes

Reviewer #2: Yes

4. Is the manuscript presented in an intelligible fashion and written in standard English?

Reviewer #1: Yes

Reviewer #2: Yes

5. Review Comments to the Author

Reviewer #1: The paper focuses on a contemporary and timely topic, namely the land transfer issues in rural China. This research area is particularly interesting in a society where the communism and capitalism exist parallel, and the question of land ownership is essential in societal terms. So, the originality of the paper is not questionable.

The abstract is well written and the title reflects the content. The introduction is appropriate, setting the topic into the context and highlights the research problem. Also, some (very few) literature review can be found here. However, no separate literature review (in fact, no literature review at all) was written in the manuscript. I recommend to establish a sound literature review where also the experiences of transition countries could be highlighted (besides the processing of the basic contemporary literature). For this purposes I recommend to include:

https://jcea.agr.hr/en/issues/article/92

https://doi.org/10.1080/08974438.2013.833567

http://agro.icm.edu.pl/agro/element/bwmeta1.element.agro-d39237c0-404d-4c36-a585-cf76c45b74a8/c/Magda_Vasa.pdf

http://ea21journal.world/index.php/ea-v168-09/

http://ea21journal.world/index.php/ea-v170-03/

The methodology is well selected and applied, I find the toolset and datasets excellent.

The results are supported by the methodology and provides us a very good understanding of the investigated problem. The conclusions are clear and acceptable.

The limitations of the research are not highlighted.

Reviewer #2: The topic of the paper is relevant and very interesting. Authors used a wide range of international literature sources and cited them correctly. Most part of them are from the last few years. The authors highlighted the importance of measures to improve the distribution of land resources and achieve moderation in China. With their research, they proved that land transfer plays an irreplaceable role in fully validating the social and economic value of land. They used official data from China Family Panel Studies, to which matching methodology was applied.

They illustrated results by using clear tables, but in some cases probably figures would have been more illustrative. I appreciate their findings and results.

Suggestions:

1. Please clarify the following sentence, because I could not understand numbers:

“. According to FAO's World Agricultural Census in 81 countries (representing 2/3 of the world's population and 38% of the world's land available for cultivation), about 73%, 85% and 95% of farmland is under 1, 2 and 5 hectares respectively.”

2. I think you will have to use capita instead of capital:

“…land per capital in the world…”

3. Please clarify the following sentence, because I could not understand numbers:

4. “…finally obtained the tracking data for a sample of 7,913 rural households in 2018, including 6475 households are in the land flow-out group, 7420 households are in the land flow-in group and 5982 land non-flow households.”

5. I suggest to write few sentences bellow the following subchapter:

“2.3. Variable selection”

6. PLOS authors have the option to publish the peer review history of their article (what does this mean?). If published, this will include your full peer review and any attached files.

Reviewer #1: No

Reviewer #2: No

---

## [Author Response · Author response to Decision Letter 0]

27 Sep 2022

Thank you for Editors and reviewers’ comments concerning our manuscript. Those comments are all valuable and very helpful for revising and improving our paper, as well as the important guiding significance to our researches. We have studied comments carefully and have made correction which we hope meet with approval. Revised portion are marked in red in the paper. The main corrections in the paper and the responds to the comments are as flowing: 

We carefully read the PLOS ONE style templates and edit our manuscript according to the guidelines for main body, affiliations and file naming to ensure our revised manuscript meets PLOS ONE's style requirements.

2. Ethics statement.

Our research do not involve human participants.

3. Please provide additional details regarding participant consent. 

I worked with my co-authors to revise the manuscript, all authors have read and agreed to the version of the revised manuscript.

4. Please ensure that you include a title page within your main document.

We have included a title page into the beginning of our manuscript and listed all authors and affiliations.

5. Upon re-submitting your revised manuscript, please upload your study’s minimal underlying data set as either Supporting Information files or to a stable, public repository and include the relevant URLs, DOIs, or accession numbers within your revised cover letter.

We have uploaded the minimal data set to the Dryad Digital Repository (DOIs: https: //doi.org/10.5061/dryad.m63xsj45d, URLs:https://datadryad.org/stash/share/DQWtYc7w2mNOxyRQrnaLIofTXXCWGyGhKNhwSE4szUs), and included in Supporting Information files. 

6. We note you have included a table to which you do not refer in the text of your manuscript. Please ensure that you refer to Tables 4& 5 in your text; if accepted, production will need this reference to link the reader to the Table.

We are sorry for the omission of Tables 4 & 5 in the article and we have clarified the placement of in our text.

Figures 1-3 in the article and provided separate captions for each figure in the manuscript

7. Please ensure that you refer to Figure xxxxx in your text as, if accepted, production will need this reference to link the reader to the figure

We are sorry for the omission of the figure Information and we have referred to the placement of Figures 1-3 in the article of in our text.

8. Please include a separate caption for each figure in your manuscript.

We have provided separate captions for each figure in the manuscript.

In the revised draft, we supplement the references that can support our findings. We make carefully check of the reference list to ensure that it is complete and correct.

The added references are as follows:

6. Liangzhen Zang. Farmland Fragmentation and Collective Action: A Study on the Irrigation System in China. 1st ed. Beijing: Economic Science Press; 2021.

26. Vasa L . BEHAVIOUR PATTERNS OF FARM-MANAGING HOUSEHOLDS AFTER THE RESTRUCTURING OF AGRICULTURE – A SOCIO-ECONOMIC ANALYSIS[J]. Journal of Central European Agriculture, 2003, 3(4). https://jcea.agr.hr/en/issues/article/92

31. Vasa L, Angeloska A, TrendovN. Comparative analysis of circular agriculture development in selected Western Balkan countries based on sustainable performance indicators. Economic Annals-XXI. 2017;168(11-12), 44-47. https://doi.org/10.21003/ea.V168-09

Response to Reviewer #1:

Thank you for your careful and thoughtful examination of our paper. Incorporating your comments, as well as those of the Editor, and the other referee, resulted in a greatly improved manuscript. We address each of your comments below.

1. The reviewer’s comment: “I recommend to establish a sound literature review where also the experiences of transition countries could be highlighted (besides the processing of the basic contemporary literature.”

Response：We greatly appreciate your comments and suggestions. We have carefully read the references you provided and absorbed the excellent information. We have rearranged our introduction, reorganized the articles cited in this paper and in the article you provided to make them clearer. We have Based on the reviewers' concerns, we have added the following refined literature review. Based on this we have added and improved the literature review. See the literature review section of the article for details please.

2. The reviewer’s comment: “The limitations of the research are not highlighted.”

Response：We are grateful for the suggestion. After careful consideration, we have added the limitations of our research as the following:

There are numerous studies on land transfer on household income distribution, due to the limitation of data availability, this paper only uses CFPS 2018 data, although the sample coverage is wide and highly representative, but it does not use panel data to study the impact of land transfer on household income, which cannot reflect the long-term variability of farm household income, which is also the direction of future research for long-term tracking of farm household income changes.

Response to Reviewer #2:

1. The reviewer’s comment: “Please clarify the following sentence, because I could not understand numbers”

“. According to FAO's World Agricultural Census in 81 countries (representing 2/3 of the world's population and 38% of the world's land available for cultivation), about 73%, 85% and 95% of farmland is under 1, 2 and 5 hectares respectively.”

Response： Here we would like to use this data set to present the state of land fragmentation. We have already explained and expanded on the above sentence as follows:

According to the FAO World Agricultural Census, 73%, 85% and 95% of the world's 81 countries (of which these 81 countries account for 2/3 of the world's population and 38% of the world's cultivable land) have farmland of less than 1, 2 and 5 hectares, respectively, which fully indicates the high degree of fragmentation of the world's farmland. In terms of distribution, small farmers are common in Europe, OECD countries and developing countries such as Brazil, India and China. This degree of farmland fragmentation seriously affects the efficiency of land use and prevents the achievement of large-scale, mechanized and intensive land management, which in turn reduces the efficiency of land resource use.

2. Response to comment: I think you will have to use capita instead of capital:

“…land per capital in the world…”

Response：Thank you very much for your suggestion, we have used capita instead of capital in our manuscript.

3. Response to comment: Please clarify the following sentence, because I could not understand numbers: “…finally obtained the tracking data for a sample of 7,913 rural households in 2018, including 6475 households are in the land flow-out group, 7420 households are in the land flow-in group and 5982 land non-flow households.”

Response：Thank you very much for your comment. Among the 7919 farm households selected here, 5982 households did not participate in land transfer. For the convenience of the analysis, we grouped the non-participating households in the land flow-out group and the land flow-in group at the same time, so that the sample size of the land transfer group is 6475 households and the land transfer group is 7420 households. the original text was modified as follows.

…finally obtained the tracking data for a sample of 7,913 rural households in 2018, including 6475 households are in the land flow-out group (Including 5982 non-flow households ), 7420 households are in the land flow-in group (Including 5982 non-flow households ).

4. Response to comment: I suggest to write few sentences bellow the following subchapter:

“2.3. Variable selection”

Response：We are grateful for the suggestion. We have modified the article accordingly, as follows:

Reasonable variable selection plays an important role in improving the robustness and universal applicability of research results. In this paper, the actual situation of sample households is fully considered in the selection of variables, and the study is mainly conducted in terms of the impact of farmland transfer on household income, which is representative in the selection of variables. 

Finally, we would like to express our gratitude to the editor and all the reviewers for the extremely helpful comments and for your guidance in the revision. We tried our best to improve the manuscript and made some changes in the manuscript. These changes will not influence the content and framework of the paper. We hope that our efforts have succeeded in allaying your concerns. We look forward to learning about your decision. And we express our thanks again to the referee for his time and efforts in reviewing our paper.

---

## [Editor Report · Decision Letter 1]

10 Oct 2022

A study of the impact of land transfer decisions on household income in rural China

PONE-D-22-16858R1

Dear Dr. Wang,

We’re pleased to inform you that your manuscript has been judged scientifically suitable for publication and will be formally accepted for publication once it meets all outstanding technical requirements.

Kind regards,

László Vasa, PhD

Academic Editor

PLOS ONE
---

## [Editor Report · Acceptance letter]

12 Oct 2022

PONE-D-22-16858R1 

A study of the impact of land transfer decisions on household income in rural China 

Dear Dr. Wang:

I'm pleased to inform you that your manuscript has been deemed suitable for publication in PLOS ONE. Congratulations! Your manuscript is now with our production department. 

Kind regards, 

on behalf of

Prof. Dr. László Vasa 

Academic Editor

PLOS ONE